# Parameter Efficient Fine-Tuning of Large Vision Foundational Models for Multi-Channel Medical Image Segmentation

**Kaushik Dutta**[1]                                          KAUSHIK.DUTTA@BMS.COM
**Devansh Agarwal**[1]                                  DEVANSH.AGARWAL@BMS.COM
**David Paulucci**[1]                                      DAVID.PAULUCCI@BMS.COM
**Angshu Rai**[1]                                              ANGSHU.RAI@BMS.COM
**Mariann Micsinai-Balan**[1]                  MARIANN.MICSINAI@BMS.COM
[1] *Bristol-Myers Squibb Company, Princeton, NJ, USA*

**Editors:** Accepted for publication at MIDL 2025

## Abstract

Multi-channel data in medial imaging where each modality encodes distinct and complementary information is critical for accurate 3D segmentation. The UNetR architecture has demonstrated success in 3D medical image segmentation by integrating transformer-based encoder with a convolutional decoder. However, full fine-tuning of UNetR for new multi-channel tasks is computationally expensive and prone to over-fitting, especially with limited data and large transformer backbones. Moreover conventional transformer models, such as Vision Transformers are typically pre-trained on single channel images, limiting their direct applicability in multi-modal imaging tasks. To address this, we propose a parameter-efficient fine-tuning strategy using channel-wise Low-Rank Adaptation adapters within the UNetR encoder framework, enabling scalable multi-channel adaptation with reduced parameter overhead.

**Keywords:** Medical Image Segmentation, Vision Transformers (ViT), UNetR, Low-Rank Adaptation (LoRA), Multi-Channel Data, Finetuning

## 1. Introduction

Medical image segmentation is increasingly leveraging large pre-trained foundation models, such as Vision Transformers (ViTs) (Dosovitskiy et al., 2020), for their powerful learned representations (Zhang et al., 2024). However, applying these pre-trained models to multi-channel volumetric medical data (e.g., multi-sequence MRI or PET/CT) is challenging, as conventional ViTs is designed to handle 2D images with fixed channel structures. Adapting transformers to effectively ingest multi-modal medical imaging data typically requires significant architectural modifications and substantial computational resources necessitating the need for parameter-efficient fine-tuning (PEFT) methods that enable large models to be adapted with only minimal changes. In this work we have explored low-rank adaptation (LoRA) based adapters for pre-trained ViT based UNetR framework in context of multi-channel medical image segmentation.

## 2. Methodology

### 2.1. Datasets and Preprocessing

We utilized two multi-channel datasets from the MedDecathlon challenge (Antonelli et al., 2022): ProstateX and BraTS. ProstateX includes 48 multi-parametric MRI studies (T2-weighted and ADC maps) with annotations for peripheral (PZ) and transition zones (TZ). BraTS comprises 750 MRI scans with four modalities (T1, T1-Gd, T2, FLAIR) from glioma patients with labels for whole tumor (WT), tumor core (TC), and enhancing tumor (ET).

### 2.2. Model Implementation

We implemented the UNetR framework, combining a pre-trained ViT-Base encoder with a CNN-based decoder for effective 3D medical image segmentation (Hatamizadeh et al., 2021). The ViT was pre-trained in a self-supervised mechanism using a masked auto-encoder (He et al., 2022) approach on an internal dataset of 0.5 million MR and CT image volumes. We introduced rank-stabilized, channel-wise LoRA adapters into the UNetR transformer

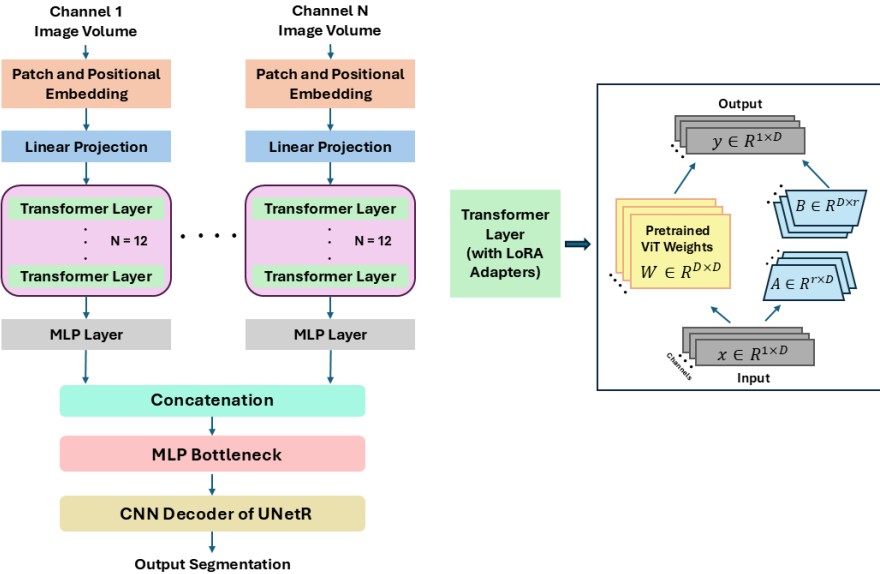

Figure 1: Architecture of our proposed multi-channel framework of our pre-trained ViT model with LoRA adapters.

encoder, by injecting low-rank trainable matrices specific to each image modality(Hu et al., 2021). Furthermore, a dynamic adapter-switching mechanism was implemented to alternate between modality-specific adapters during training and inference, facilitating efficient multi-modal feature extraction. The resulting latent representations from each LoRA adapters are concatenated and passed through an MLP block to reduce dimensionality before CNN-decoding block.

### 2.3. Implementation Details

We trained and evaluated the UNetR model under four configurations: (1) training from scratch (Baseline), (2) full fine-tuning with pre-trained ViT weights (Finetune), (3) decoder-only fine-tuning with a frozen encoder (Frozen), and (4) our proposed method using channel-wise LoRA adapters (LoRA). We evaluated the effect of LoRA rank *(r=16, 32, 64)* on performance and parameter efficiency. All models were trained for 200 epochs, using Adam optimizer with OneCycleLR scheduler (max LR = 0.005) and optimized using a combined Dice and cross-entropy loss. Model performance for segmentation was assessed using Dice Score thresholded at 0.5.

## 3. Results & Conclusion

The experimental results from different configurations are summarized in Table 1. UNetR-LoRA achieves performance comparable to full fine-tuning, while significantly reducing the number of trainable parameters (Table 2) and outperforming decoder-only and baseline configurations.

Table 1: Performance Comparison with Percentage Changes w.r.t. Baseline. Best values in bold, second-best underlined. Arrows indicate improvement ($\uparrow$) or drop ($\downarrow$).

| | Region | Baseline | Finetune | Frozen | LoRA-16 | LoRA-32 | LoRA-64 |
|---|---|---|---|---|---|---|---|
| | TC | 0.8325 | **0.8482** ($\uparrow$1.9) | 0.8149 ($\downarrow$2.1) | 0.8284 ($\downarrow$0.5) | 0.8376 ($\uparrow$0.6) | 0.8128 ($\downarrow$2.4) |
| BRaTS | WT | 0.8873 | **0.8998** ($\uparrow$1.4) | 0.8932 ($\uparrow$0.7) | 0.8918 ($\uparrow$0.5) | 0.8941 ($\uparrow$0.8) | 0.8894 ($\uparrow$0.2) |
| | ET | 0.6593 | **0.6651** ($\uparrow$0.9) | 0.6342 ($\downarrow$3.8) | 0.6482 ($\downarrow$1.7) | 0.6491 ($\downarrow$1.5) | 0.6316 ($\downarrow$4.2) |
| | Overall | 0.8546 | **0.8693** ($\uparrow$1.7) | 0.8562 ($\uparrow$0.2) | 0.8564 ($\uparrow$0.2) | 0.8602 ($\uparrow$0.7) | 0.8499 ($\downarrow$0.5) |
| | PZ | 0.5412 | **0.5958** ($\uparrow$10.1) | 0.5284 ($\downarrow$2.4) | 0.5645 ($\uparrow$4.3) | 0.5877 ($\uparrow$8.6) | 0.5669 ($\uparrow$4.7) |
| ProstateX | TZ | 0.7226 | **0.7364** ($\uparrow$1.9) | 0.7164 ($\downarrow$0.9) | 0.6967 ($\downarrow$3.6) | 0.7167 ($\downarrow$0.8) | 0.7314 ($\uparrow$1.2) |
| | Overall | 0.6369 | **0.6661** ($\uparrow$4.6) | 0.6224 ($\downarrow$2.3) | 0.6452 ($\uparrow$1.3) | 0.6522 ($\uparrow$2.4) | 0.6492 ($\uparrow$1.9) |

Table 2: Parameter Comparison of different UNetR configurations

| | | Baseline | Finetune | Frozen | LoRA-16 | LoRA-32 | LoRA-64 |
|---|---|---|---|---|---|---|---|
| | Total | 142.89 M | 142.9 M | 142.9 M | 150.13 M | 157.35 M | 171.81 M |
| Params | Train. | 142.89 M | 142.9 M | 54.58 M | 58.71 M | 62.84 M | 71.09 M |
| | | (100.0%) | (100.0%) | (38.16%) | (39.09%) | (39.93%) | (41.33%) |
| | Non-Train. | 0 M | 0 M | 88.32 M | 91.42 M | 94.52 M | 100.72 M |

## 4. Conclusion

This paper proposes integrating channel-wise LoRA adapters into the transformer encoder of the UNetR framework to enable efficient multi-channel image segmentation. Experimental results demonstrate that the LoRA-based approach outperforms baseline and frozen encoder configurations, while achieving performance comparable to full fine-tuning with a 40% reduction in trainable parameters—highlighting its suitability for leveraging pre-trained models in downstream medical imaging tasks.

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
