# OpenReview forum: "Parameter Efficient Fine-Tuning of Large Vision Foundational Models for Multi-Channel Medical Image Segmentation"
_MIDL.io/2025/Short_Papers — MIDL 2025 - Short Papers_

### Official Review · Reviewer_zyYA · 2025-04-24

**Rating:** 4
**Confidence:** 4

**Summary:**

The paper introduces a parameter-efficient fine-tuning strategy using channel-wise LoRA adapters within the UNetR framework for multi-channel 3D medical image segmentation. It leverages a ViT encoder pre-trained with masked auto-encoders and injects modality-specific adapters to reduce training overhead. Tested on ProstateX and BraTS datasets, the method achieves performance close to full fine-tuning while reducing trainable parameters by about 40%.

**Strengths:**

- Tackles the problem of adapting single-channel pre-trained vision transformers to multi-channel medical imaging.
- Introduces channel-wise LoRA adapters, reducing trainable parameters substantially.
- Performance closely matches full fine-tuning while being significantly more efficient.
- Implements a modality-specific adapter-switching mechanism, enhancing flexibility in multi-modal settings.
- Evaluated on two public datasets with detailed comparison across multiple configurations and LoRA ranks.

**Weaknesses:**

- Gains in segmentation performance over decoder-only or baseline are marginal in many cases.
- Limited discussion on the overhead introduced by additional parameters from LoRA adapters.
- No ablation to isolate the impact of adapter-switching or modality-wise architecture choices.

---

### Decision · Program_Chairs · 2025-05-01

Accept